# Risk Factors for Fall-Related Serious Injury among Korean Adults: A Cross-Sectional Retrospective Analysis

**DOI:** 10.3390/ijerph18031239

**Published:** 2021-01-30

**Authors:** Moon-Sook Kim, Hyun-Myung Jung, Hyo-Yeon Lee, Jinhyun Kim

**Affiliations:** 1Medical Nursing Department, Seoul National University Hospital, Seoul 03080, Korea; behappynow@snuh.org (M.-S.K.); wise8431@hanmail.net (H.-M.J.); 2Infection Control Center, Seoul National University Hospital, Seoul 03080, Korea; dryiyoun@snuh.org; 3College of Nursing, Seoul National University, Seoul 03080, Korea

**Keywords:** falls, serious injury, length of stay, inpatients, risk assessment

## Abstract

The purpose of this study was to identify the risk factors of serious fall-related injuries by analyzing the differences between two fall groups: one with serious fall-related injuries and one without such injuries. Applying a retrospective, descriptive investigation study design, we analyzed the degree of fall-related injury and the risk factors related to serious falls by conducting a complete survey of the medical records of fall patients reported throughout one full year, 2017, at a tertiary hospital in Seoul, Korea. Among the patients with reported falls, 188 sustained no injury (63.1%), 72 sustained minor injury (24.2%), and 38 patients sustained serious injury (12.8%). The serious fall-related injuries included eight lacerations requiring suture (2.7%), 23 fractures (7.7%), five brain injuries (1.7%), and two deaths (0.7%). Analysis results indicated that taking anticoagulants/antiplatelet drugs (*p* = 0.016) and having a fall history (*p* = 0.038) were statistically significant in the differences between the group with serious injury related to falls and the group without serious injury. Logistic regression revealed that taking anticoagulant/antiplatelet drugs was the factor most significantly correlated with serious injuries related to falls (OR = 2.299, *p* = 0.022). Results show that it is necessary to develop a patient-tailored fall prevention activity program.

## 1. Introduction

Falls are not common among young people. Fall-related injuries occur in approximately 25% to 55% of patients admitted to hospitals that care for older adults in Korea, and 1.2% to 16.2% of falls involve serious injuries such as fractures, cerebral hemorrhage, and death [1,2,3]. Fall-related injuries lead to extended hospital stays and increased medical expenses [4], and may lead to complaints such as mental distress and medical litigation for patients and their families [5]. From a perspective of nursing workforce policy, falls as patient outcome are the main issue in the performance of nurse staffing policy.

To reduce fall-related injuries, it is necessary to predict the incidence of falls and to perform fall prevention activities that consider the characteristics of the patient [6]. Nurses routinely use a fall risk assessment tool (e.g., Morse Fall Scale), which has been verified for reliability and validity, and perform fall-prevention activities according to the evaluation results. However, the number of reported falls and injuries related to falls continues to increase, even in patients determined to have a low fall risk [3,7,8].

In recent years, studies have been conducted to clarify the risk factors related to falls; age, fracture risk, use of anticoagulants, surgery during hospitalization, male, history of falling, a history of joint replacement surgery, and use of antipsychotics have been reported [1]. Other fall risk assessment instruments include the Ability-specific Balance Competence Scale (ABCS), which identifies the risk factors of age (>85), bone/orthopedic conditions, anti-coagulation, and surgery during hospitalization, and the St. Thomas Risk Assessment Tool in Falling Elderly Inpatients (STRATIFY) examines excitement, visual impairment, frequent toilet access, and walking ability. The fracture risk assessment tool (FRAX^TM^) is available to evaluate the risk factors of age, previous fracture, parents’ history of hip fracture, smoking, steroid use, heavy drinking, and rheumatoid arthritis. Using various tools, or using them together, is recommended [5], but the predictive power of combining such tools to evaluate a patient’s risk for fall-related injury has not been verified. In Korea, studies have shown that advanced age, requiring the emergency room, and non-use of ambulatory aid are related to fall-related injuries [9], but studies analyzing various risk factors of serious injuries related to falls are insufficient. Therefore, in order to reduce the number and seriousness of injuries related to falls in hospitalized patients in a tertiary hospital, it is necessary to identify the risk factors for serious injuries related to falls that are applicable to domestic conditions, and to develop and apply a serious injury assessment tool for falls that reflects the risk factors.

The purpose of this study was to use clinical data of patients reported as falls to determine the degree of fall-related injuries and to identify risk factors for serious injuries related to falls as evidence for interventions to prevent serious injuries related to falls. We identified factors that should be examined in addition to the limited scope of the Morse assessment to differentiate risks for serious injury from falls in hospitals.

## 2. Materials and Methods

### 2.1. Research Design

This study was a retrospective, descriptive investigation that analyzed the degree of fall-related injury and the risk factors related to serious fall injuries among hospitalized patients.

### 2.2. Participants and Data Collection

The participants of this study were 298 patients who met the inclusion criteria among fall patients aged 19 years or older among hospitalized patients from January to December 2017. Exclusion criteria were children’s ward, psychiatric ward, intensive care unit, emergency room, and outpatient fall patients, and those who were not evaluated on the Morse Fall Scale.

Using a case study form developed by this research team, the data were collected from electronic medical records and fall reports. The case study form is divided into the patient’s general characteristics, fall-related characteristics, and clinical characteristics. General characteristics consist of age, sex, and length of stay. Fall-related characteristics consist of the existence or absence of serious injury, degree of injury related to falls, type of injury related to falls, caregiver presence, history of falling, intravenous therapy at the time of fall, use of ambulatory aid, degree of fall risk, and existence or absence of environmental factors. Clinical characteristics include smoker, drinker, urinary frequency, visual impairment, cognitive impairment, lower extremities, weakness, osteoporosis, bone metastases/bone tumor, history of fracture, surgery during hospitalization, rheumatoid arthritis, taking anticoagulant/antiplatelet drugs, taking antipsychotic drugs, taking steroids, and having a bleeding tendency.

To minimize errors and reduce deviations in the data, three of the researchers cross-checked the information.

### 2.3. Ethics

In the interest of protecting the research subjects, the study design received approval (1805-134-948) from the Ethics Committee of Seoul National University Hospital, and then we received approval to access medical records. Data collected through medical records were managed such that no one other than the three designated researchers could access them. Each participant’s personal information was encrypted using symbols, and an encrypted computer was used to analyze the research data. Documented research data were locked in a secure location and all of the information is to be discarded three years after the end of the study.

### 2.4. Definition of Variables

#### 2.4.1. Falls

A fall is defined as a sudden drop to the floor or a place lower than the current position due to an unintended posture change; for the purposes of this study, it does not include sudden falls due to a stroke or fainting or falls due to a strong external force (wind, being pushed by others, etc.) [10].

#### 2.4.2. Degree of Injuries Related to Falls

In this study, data were collected by dividing fall-related injuries into five stages: no harm, minor injury, moderate injury, severe injury, and death. No harm means that no damage occurred to the patient. Minor injury means that the patient was injured but required additional observation, medication, or primary treatment. Moderate injury refers to a case that included an unplanned extended hospital stay, surgery, treatment, and so on. Severe injury refers to a case of permanent injury such as brain injury or disability. Death refers to a case of death as a direct result of a fall [11].

Based on previous studies, in this study, if the collected data indicated no harm or minor injury, the fall was recorded as being without serious injury, and if the collected data indicated moderate injury, severe injury, or death, the fall was recorded as having serious injury [12].

#### 2.4.3. Surgery during Hospitalization

Surgery during hospitalization refers to a case where surgery was performed during hospitalization before the fall, and was not related to the fall.

#### 2.4.4. History of Fracture

A participant’s history of fracture was defined as having experienced a fracture (regardless of cause) before the occurrence of the fall as determined through a review of the medical records.

#### 2.4.5. Bleeding Tendency

Bleeding tendency was defined, through consultation of an expert group, as those with a glomerular filtration rate of 50 or less, prothrombin time of 50% or less, platelets of 50,000 or less, or blood coagulation disorders.

### 2.5. Data Analysis

The collected data was statistically analyzed as follows: The degree of injury related to falls was analyzed using descriptive statistics.The t-test and χ2 test were performed to determine the differences between the group with serious fall-related injury and the group without.Logistic regression analysis was performed to determine the effects of participants’ general characteristics, fall-related characteristics, and clinical characteristics on serious fall-related injuries.

## 3. Results

### 3.1. Participants’ Fall-Related Injuries

Among the 298 patients with reported falls, 188 had no related injury (63.1%), 72 patients had minor injury (24.2%), 31 patients had moderate injury (10.4%), and 5 patients had severe injury (1.7%); 2 of the patients had fatal injuries (0.7%). Of the 38 patients with serious (moderate to fatal) injuries (12.8%), 8 involved lacerations requiring sutures (2.7%), 23 involved fractures (7.7%), and 5 had brain injuries (1.7%), whereas 2 falls led to the patients’ deaths (0.7%) (Table 1).

### 3.2. Characteristics of Fall Patients

Table 2 describes the age, sex, and health-related characteristic data of all the patients with reported falls. The only statistically significant difference in the characteristics of fall patients between the serious injury group and the group without serious injury was taking anticoagulant/antiplatelet drugs (*p* = 0.016) (Table 2).

### 3.3. Fall-Related Characteristics

The fall-related characteristics include nurse’s shift, caregiver presence, history of falling. intravenous therapy at the time of fall, use of ambulatory aid, degree of fall risk, and environmental factors. There was no statistically significant difference in the fall-related characteristics between the serious injury group and the group without serious injury except for the history of falling (*p* = 0.038) (Table 3).

### 3.4. Factors Associated with Serious Injury Related to Falls

We included taking anticoagulants/antiplatelet drugs and fall history as independent variables of logistic regression, as they were the two statistically significant differences between the group with serious injury related to falls and the group without serious injury. Lower extremity weakness, surgery during hospitalization, caregiver presence, and environmental factors at the time of the fall were added as independent variables and analyzed. Taking anticoagulant/antiplatelet drugs was again found to be the factor that significantly affected serious injuries related to falls (*p* = 0.022) (Table 4).

## 4. Discussion

The purpose of this study was to identify the risk factors of serious fall-related injuries by analyzing the difference between the fall group with serious injuries related to falls and falls without related serious injuries. The general characteristics of the participants were compared, including age, sex, and length of stay. The mean ages of the group with serious fall-related injury and the group without serious fall-related injury were 65.9 and 63.9 years, respectively, which was not significantly different. There were 18 men (47.4%) and 20 women (52.6%) in the fall group with serious injuries related to falls, and 150 men (57.7%) and 110 women (42.3%) in the fall group without serious injuries related to falls; there was no significant difference between sexes, which is consistent with previous studies [1,13]. The fall group with serious injurious falls stayed in the hospital longer than the fall group without serious injurious falls (11.8 vs. 16.2 days), which also confirms the results of previous studies [13].

The variables of this study included many items suggested by the fracture risk assessment tool (FRAX^TM^) [5] for assessing the risk of injury related to falls (history of fracture, smoker, taking steroids, drinkers, and rheumatoid arthritis) and several variables used in a variety of previous studies (drinker, urinary frequency, visual impairment, cognitive impairment, lower extremity weakness, osteoporosis, bone metastases/bone tumors, surgery during hospitalization, taking antipsychotic drugs, bleeding tendency, and taking anticoagulant/antiplatelet drugs). Fracture risk factors, such as history of fracture, smoking, use of steroids, drinkers, rheumatoid arthritis, osteoporosis, and bone metastases/bone tumors, were not significant and were consistent with the results of Aryee et al. [1].

However, Toyabe et al. [5] recommended that STRATIFY, a fall risk assessment tool, and FRAX^TM^, a fracture risk assessment tool, be evaluated together to predict serious injury. Therefore, it is believed that it is difficult to predict the risk of serious injury related to falls using the Morse Fall Scale, which is currently in use as the standard assessment tool in hospital settings.

After consulting an expert in each department, we determined the criteria for the bleeding tendency as patients having a glomerular filtration rate of 50 or less, prothrombin time of 50% or less, platelets of 50,000 or less, and blood coagulation disorder. The bleeding tendency was slightly higher in the fall group with serious fall-related injuries, but there was no significant difference (28.9 vs. 22.7%). This was consistent with the results of previous studies that reported that the bleeding tendency was not associated with fall-related injury [13].

Anticoagulant therapy was examined in combination with anticoagulant and antiplatelet drugs. There was a significant difference in the fall group of 18 patients (47.4%) with serious fall-related injury and 73 fall patients (28.1%) without serious fall-related injury. As a result, patients taking anticoagulants or antiplatelet drugs had a 2.299 times higher incidence of serious fall-related injury than those who did not, which can be considered a predictor of serious fall-related injury. This was different from the study by Aryee et al. [1], which reported that taking anticoagulants could not predict the risk of fall-related injury [1]. This result suggests that medical staff should be particularly diligent regarding fall prevention for patients taking anticoagulants or antiplatelet drugs.

Three patients (7.8%) in the fall group with related serious injuries had a surgery during hospitalization compared to 43 patients (16.5%) in the group without fall-related serious injuries. Falls present a risk of opening the surgical site if the sutures have not yet been removed following major surgery on the abdomen, chest, and lower extremities [14]. Fortunately, in the reported cases, the body areas injured by the falls were not their surgical sites.

Taking antipsychotics was not shown to make a significant difference with or without serious injury (31.6% vs. 31.2%), which is inconsistent with the results of a study by Mion et al. [15], who found that taking antipsychotics was a risk factor for fall-related injuries. Taking two or more antipsychotics was reportedly related, which we did not find, although some patients in this study were taking several drugs for conditions such as insomnia, delirium, depression, and schizophrenia. It is believed that the association was found in the earlier study because the use of antipsychotics was the only factor investigated [15].

Also included in the fracture risk assessment tool, we investigated whether taking steroids was associated with a risk of fracture due to the nature of the drug. Twelve patients (31.6%) in the fall group had serious injuries related to falls, and 65 (25%) in the fall group without serious injury related to the fall. There was no difference between the groups, but a previous study showed that in cancer patients, taking steroids is a risk factor for fall-related injury [12].

As for the serious injuries related to falls, understanding the fall situation was considered important, and as the characteristics of the fall, the caregiver presence, intravenous therapy at the time of fall, the risk degree of fall, the use of ambulatory aid, the environmental factor, and the history of falling were investigated. In the fall group with serious injuries related to falls, 17 (44.7%) were in the presence of a caregiver at the time of the fall, and 80 (30.8%) in the fall group without serious injuries related to the fall. If there was no caregiver, the incidence of serious injury was higher (30.8% vs. 44.7%), but there was no significant difference. Krauss et al. [16] reported that serious injury occurs when no one is there as support at the time of the fall, which is not consistent with the results of this study. This is thought to be due to the lack of understanding of the role of the caregiver in preventing falls and injury, although education is being given to residents to prevent falls. Therefore, it is thought that the participation of the caregiver in fall prevention activities in the future can effectively prevent fall-related injuries.

The history of falling was 11 (28.9%) in the fall group with serious injuries related to falls, and 40 (15.4%) in the fall group without serious injuries related to falls. In the fall group with serious injuries related to falls, there were many cases with a history of falling, and a significant difference was found compared to the fall group without related serious injury, which is consistent with the results of previous studies [1]. Therefore, patients who have previously experienced a fall may be more likely to suffer a serious injury after a fall, highlighting the need for fall prevention. In this study, there were 11 cases of patients with repeated falls: One patient reported four falls, one reported three, and nine reported two. Six of them were found to have suffered fall-related injuries such as skull fractures, bruises, abrasions, and bleeding gums.

There was no significant difference between the two groups in the use of ambulatory aids such as wheelchairs and walkers (18.4% vs. 17.3%). This result conflicts with that of another study that reported non-use of the ambulatory aid was related to fall-related injuries [9].

Based on the results of this study, fall prevention education should be offered to patients taking anticoagulant/antiplatelet drugs to prevent serious injury related to falls. In addition, it is thought that patients with a high probability of falls can be provided with customized education programs that can prevent falls and minimize fall-related injuries.

The limitation of this study is that it cannot be generalized because the data were analyzed for one year on reported falls and adult patients in a hospital.

## 5. Conclusions

Falls in hospitalized patients can cause serious injuries such as fractures, cerebral hemorrhage, and death. In addition, fall-related injuries lead to extended hospital stays and increased medical expenses. They can also cause misery for the patients and their families and can lead to civil complaints, such as medical lawsuits. Therefore, it is necessary to have a means of predicting the risk of serious fall-related injuries and provide preventive interventions. In this study, we compared the characteristics between groups with and without serious fall-related injuries to identify the risk factors for serious fall-related injuries.

We conducted a cross-sectional retrospective descriptive research study analyzing the medical records of adult patients over 19 years of age with reported falls at the Seoul National University Hospital from January to December 2017. Anticoagulant/antiplatelet drugs were found to be predictors of serious fall-related injuries. These results suggest that patients taking anticoagulant/antiplatelet drugs should be designated as a high-risk group for serious injuries related to falls requiring active interventions for injury prevention.

Hospitals use a fall risk assessment tool that has proven validity and reliability, yet it is inadequate for predicting falls and serious fall-related injuries. To predict falls and injuries related to falls, hospitals must include evaluations of diverse characteristics and factors to determine the degree of risk of falls and injury in addition to the basic fall risk assessment tool. The study of a fall risk will be a basis to develop the measurement of nursing care needs in hospitals.

## Figures and Tables

**Table 1 ijerph-18-01239-t001:** Consequences of fall (N = 298).

Characteristics	Classification	*n* (%)
Fall	Without serious injury	260 (87.2)
	With serious injury	38 (12.8)
Degree of injury related to falls	None	188 (63.1)
	Mild	72 (24.2)
	Moderate	31 (10.4)
	Severe	5 (1.7)
	Death	2 (0.7)
Type of injury related to falls	Brain injury	5 (1.7)
	Fracture	23 (7.7)
	Laceration	8 (2.7)
	Abrasion	49 (16.4)
	Bruise	43 (14.4)

**Table 2 ijerph-18-01239-t002:** Characteristics of patients with reported falls.

Characteristics		Total (N = 298)	With Serious Injuries (N = 38)	Without Serious Injuries (N = 260)	*p*
		*n* (%)	*n* (%)	*n* (%)	
Age	≤54	69 (23.2)	6 (15.8)	63 (24.2)	0.337
	55–64	69 (23.2)	10 (26.3)	59 (22.7)	
	65–74	80 (26.8)	14 (36.8)	66 (25.4)	
	≥75	80 (26.8)	8 (21.1)	72 (27.7)	
Sex	Female	130 (43.6)	20 (52.6)	110 (42.3)	0.231
	Male	168 (56.4)	18 (47.4)	150 (57.7)	
Length of stay (day, mean ± SD)		15.66 ± 20.33	11.81 ± 10.68	16.22 ± 21.33	0.212
Smoker (y)		39 (13.1)	5 (13.2)	34 (13.1)	0.999
Drinker (y)		31 (10.4)	6 (15.8)	25 (9.6)	0.255
Urinary frequency (y)		32 (10.7)	5 (13.2)	27 (10.4)	0.578
Visual impairment (y)		20 (6.7)	3 (7.9)	17 (6.5)	0.729
Cognitive impairment (y)		59 (19.8)	5 (13.2)	54 (20.8)	0.271
Lower extremities weakness (y)		101 (33.9)	17 (44.7)	84 (32.3)	0.131
Osteoporosis (y)		12 (4.0)	2 (5.3)	10 (3.8)	0.678
Bone metastases/bone tumor (y)		39 (13.1)	6 (15.8)	33 (12.7)	0.607
History of fracture (y)		26 (8.7)	4 (10.5)	22 (8.5)	0.757
Surgery during hospitalization (y)		46 (15.4)	3 (7.9)	43 (16.5)	0.168
Rheumatoid arthritis (y)		1 (0.3)	0 (0.0)	1 (0.4)	0.999
Taking anticoagulant/antiplatelet drugs (y)		91 (30.5)	18 (47.4)	73 (28.1)	0.016
Taking antipsychotic drugs (y)		93 (31.2)	12 (31.6)	81 (31.2)	0.999
Taking steroids (y)		77 (25.8)	12 (31.6)	65 (25.0)	0.387
Bleeding tendency (y)		70 (23.5)	11 (28.9)	59 (22.7)	0.396

SD, standard deviation; y, yes.

**Table 3 ijerph-18-01239-t003:** Fall-related characteristics of patients.

Characteristics		Total (N = 298)	With Serious Injuries (N = 38)	Without Serious Injuries (N = 260)	*p*
		*n* (%)	*n* (%)	*n* (%)	
Nurse’s shift	Day	97 (32.6)	12 (31.6)	85 (32.7)	0.942
	Evening	83 (27.9)	10 (26.3)	73 (28.1)	
	Night	118 (39.6)	16 (42.1)	102 (39.2)	
Caregiver presence	Yes	97 (32.6)	17 (44.7)	80 (30.8)	0.092
History of falling	Yes	51 (17.1)	11 (28.9)	40 (15.4)	0.038
Intravenous therapy at the time of fall	Yes	156 (52.3)	21 (55.3)	135 (54.9)	0.700
Use of ambulatory aid	Yes	52 (17.4)	7 (18.4)	45 (17.3)	0.874
Degree of fall risk ^a^	No	94 (31.5)	13 (34.2)	81 (31.2)	0.821
	Low	121 (40.6)	16 (42.1)	105 (40.4)	
	High	83 (27.9)	9 (23.7)	74 (28.5)	
Environmental factor	Yes	57 (19.1)	11 (28.9)	46 (17.7)	0.099

^a^ Morse Fall Scale: no (0–24), low (25–44), high (≥45); SD, standard deviation.

**Table 4 ijerph-18-01239-t004:** Factors associated with serious injury related to falls.

Variables	Coeff	SE	Wald	*p*	OR	95% CI	
						Lower	Upper
Lower extremities weakness	0.391	0.377	1.079	0.299	1.479	0.707	3.095
Surgery during hospitalization	−0.691	0.646	1.146	0.284	0.501	0.141	1.776
Taking anticoagulant/ antiplatelet drugs	0.833	0.364	5.222	0.022	2.299	1.126	4.696
Caregiver presence	0.563	0.367	2.359	0.125	1.756	0.856	3.603
History of falling	0.641	0.417	2.364	0.124	1.899	0.839	4.299
Environmental factor	0.378	0.206	3.361	0.067	1.460	0.974	2.187

Coeff = coefficient; CI = confidence interval; OR = odds ratio; Hosmer–Lemeshow (goodness-of-fit test) χ2 = 8.015, *p*=0.331; Nagelkerke R^2^ = 0.106, correct classification (%) = 86.8%.

## Data Availability

Data sharing not applicable.

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
