# Peer review of "Risk Factors for Fall-Related Serious Injury among Korean Adults: A Cross-Sectional Retrospective Analysis"

_ijerph, 2021, doi:10.3390/ijerph18031239_

Round 1
Reviewer 1 Report
Please find the attached file. Thanks!

Author Response
Thank you for the comments and helpful suggestions regarding our manuscript.Please check the attached file.

Reviewer 2 Report
The paper is a descriptive analysis of fall-related injuries from patients at a tertiary hospital, comparing risk factors between minor and serious injuries resulting from such falls. The authors are sufficiently thorough and clear in their description. The authors considered the ethics of accessing patient data and the research was appropriately reviewed. This paper will add to the literature on risk factors for severe falls among hospitalized patients.
A few comments for the authors' consideration:
A reader may not initially understand that the falls occurred during hospitalization. Please clarify this in the title, abstract, and the methods (line 73-74). I honestly thought that this paper was about falls that caused the initial hospitalization and it wasn't clear until discussion (line 227) where the authors write "falls in hospitalized patients..."
Is it possible to provide more detail about the circumstances of the patients' falls in order to inform future prevention methods? I'm thinking of person, place, and time. The authors include "Environment status at the time of fall" within Table 3. Is there information about the location of the fall, e.g., in/near toilet, shower, patient bed, etc.? Is there information about the time of the fall? For example, were patients more likely to fall during the early morning hours or during a nurse shift-change, etc.
The authors should consider including more in the discussion/conclusions about how their results might inform prevention efforts, including suggested interventions. Some of the authors' references may include related interventions, such as references 6 and 7.
Minor errors:
line 77: missing punctuation after "report"
lines 194-195: there may be a word missing after "steroids"
line 203: please clarify "CG" as this is not described elsewhere
Author Response

(The authors gave the same response as above.)
